# The Impact of Temperature on Host–Parasite Interactions and Metabolomic Profiles in the Marine Diatom *Coscinodiscus granii*

**DOI:** 10.3390/plants13233415

**Published:** 2024-12-05

**Authors:** Ruchicka Annie O’Niel, Georg Pohnert, Marine Vallet

**Affiliations:** 1Institute for Inorganic and Analytical Chemistry, Friedrich Schiller University Jena, 07743 Jena, Germany; ruchicka.oniel@uni-jena.de (R.A.O.); georg.pohnert@uni-jena.de (G.P.); 2Max Planck Fellow Group Plankton Community Interaction, Max Planck Institute for Chemical Ecology, 07745 Jena, Germany

**Keywords:** bloom-forming algae, diatoms, parasitoid oomycete, *Lagenisma coscinodisci*, host–parasite interactions, UHPLC-HRMS, untargeted metabolomics, temperature increase

## Abstract

Diatoms are single-celled photosynthetic eukaryotes responsible for CO_2_ fixation and primary production in aquatic ecosystems. The cosmopolitan marine diatom *Coscinodiscus granii* can form seasonal blooms in coastal areas and interact with various microorganisms, including the parasitic oomycete *Lagenisma coscinodisci*. This unicellular eukaryote is mainly present in the northern hemisphere as an obligate parasite of the genus *Coscinodiscus.* Understanding the interplay of abiotic factors such as temperature and biotic factors like parasitism on algal physiology is crucial as it dictates plankton community composition and is especially relevant during environmental changes and warming events. This study investigates the impact of two temperatures, 13 °C and 25 °C, on *Coscinodiscus granii* under laboratory conditions. A decreased infection rate of the parasite was observed at the elevated temperature. Comparative metabolomic analysis using UHPLC-HRMS revealed that temperature and parasitism significantly affect the algal cell metabolome. Abundances of metabolites related to sulfur metabolism, including cysteinoleic acid and dimethylsulfoniopropionate, as well as molecules linked to fatty acid metabolism, e.g., carnitine, acetylcarnitine, and eicosapentanoic acid, significantly increase in cells grown at a higher temperature, suggesting the enhanced rate of metabolism of host cells as the temperature rises. Our study reveals how temperature-induced metabolic changes can influence host–parasite dynamics in a changing environment.

## 1. Introduction

Phytoplankton comprises microalgae, such as diatoms and cyanobacteria, which form the basis of the marine food webs. The algae grow by interacting with diverse microorganisms, which creates the observable population dynamics of algal blooms [1]. Microalgae and their associated symbiotic microbes exchange chemical cues that regulate host physiology, metabolism, and defense [2,3,4]. The region surrounding phytoplankton cells where chemical exchanges occur is the phycosphere, which can attract many opportunists and microbial pathogens such as eukaryotic parasites [5]. Diverse microbial parasites targeting the group of diatoms and dinoflagellates can contribute to the termination of algal blooms in coastal ecosystems [6,7]. Parasites also play an essential role in bloom dynamics and ocean ecosystems [8], yet the impact of warming temperatures on their ecology and virulence remains poorly understood. The molecular patterns associated with the thermal adaptation of the parasites and the hosts need to be characterized if predictions on the future ocean system’s functioning are made [9]. Climate changes, including temperature increases, can induce complex restructuring of the plankton community, impacting metabolic activities and the productivity of aquatic microorganisms in a hitherto only incompletely understood interaction framework [10].

Environmental conditions such as temperature, pH, nutrient availability, and salinity modulate the physiology of microalgae in interactions and influence the behavior of aquatic pathogens and parasites [11]. An increasing number of studies suggest that higher temperatures cause adverse effects on the fitness of plankton parasites [12,13], with some aquatic microorganisms unable to adapt to rising temperatures [14]. Parasite presence can influence plankton composition, as some algal species can get infected while others cannot [15].

Untargeted metabolomics using MS or NMR instruments can identify metabolites within algal cells and those released into the environment. These released metabolites can be exchanged with coexisting microbes, especially within the diffusion-limited region around algal cells, the phycosphere. Alkaloids from the ß-carboline family involved in regulating the parasite infection of the bloom-forming diatoms *Coscinodiscus granii* were identified using UHPLC-HRMS analysis [16]. Furthermore, using an LDI-HRMS instrument, low-molecular-weight metabolites can be identified in diatoms and dinoflagellates infected by eukaryotic parasites, including phaeophorbide *a* and dimethylsulfoniopropionate [17]. There has been progress in using metabolomics to explore the host response to parasite infection, both in terms of how parasites derive energy from hosts and how hosts can modify their metabolism to resist infection [18]. However, no studies address the metabolic processes during host–parasite interaction during temperature increase, even if such multifactorial situations will be the rule, not the exception.

Here, we aimed to study the impact of temperature on the algal host abundance, parasite infectivity, and cellular metabolome. The algal host, *C. granii*, and the parasite *L. coscinodisci,* were isolated in Helgoland waters during the bloom of November 2019, and we aimed to test the impact of two temperatures, i.e., 13 °C and 25 °C, on the cell abundance, parasite infectivity, and cellular metabolic production. We investigated the metabolic profiles of diatom cultures exposed to different temperatures and parasite treatments. We selected the diatom species *Coscinodiscus granii*, which has a unique morphology and biotechnological potential. *C. granii* is used in studying silicification and environmental adaptation and represents a versatile and valuable model for a wide range of studies in diatom biology, ecology, and applied nanotechnology research [19]. The genus *Coscinodiscus* is also known to cause harmful algal blooms [20,21], and our goal was to study the effect of temperature on host cell abundance, parasite infectivity, and cell metabolome. We might then better understand the dynamics that might lead to blooms and how they can be controlled by natural enemies. We recorded the cell abundance and infection rates at two ecologically relevant temperatures, and we performed infection assays with the specialist parasitoid *L. coscinodisci* to test its infectivity on its host. We aim to identify and discuss metabolic alterations in the cellular metabolome of *C. granii* that occur in conjunction with temperature increase and parasite infection. This study aims to provide metabolic information on the collective impact of increased temperature and parasite infection on the physiology and cell metabolism of bloom-forming diatoms. Our objective was to provide biological control insights into parasite infectivity targeting harmful algal blooms, specifically caused by *Coscinodiscus* spp. Understanding the metabolic shifts revealed how these organisms adapt to stressors and may help predict responses to environmental changes.

## 2. Results

### 2.1. Impact of Temperature on Diatom Host Abundance and Parasite Infectivity

In a first experiment and to provide a comparative analysis of how temperature affects infectivity responses and algal cell abundance, the diatom species *Coscinodiscus granii*, a cosmopolitan and bloom-forming diatom, was cultivated at two incubation temperatures, i.e., 13 °C and 25 °C (an initial cell density of 5 cells mL^−1^). The lower temperature (13 °C) represents cooler conditions often found in certain marine environments, while 25 °C reflects warmer conditions that could be more prevalent due to environmental changes. This range helps assess the impact of temperature on the metabolic responses of *C. granii* and the infectivity of the parasitoid, providing valuable insights into potential ecological shifts under different thermal scenarios. We recorded the cell density over 29 days and highlighted that, in the early and exponential growth phase (Day 6 to Day 12), the cell density was significantly higher in cultures grown at 25 °C (Appendix A).

In a second experiment using dense cultures of *C. granii* with an initial cell density of 200 cells mL^−1^, we tested whether the parasitoid oomycete *Lagenisma coscinodisci* infects the algal cells at the two incubation temperatures. We selected this second experiment with a higher cell density for metabolic profiling and aimed to profile infected cells in the late infection stage (sporocysts), which corresponds to day 7 of incubation. *C. granii* was susceptible to *Lagenisma* infection at the two tested temperatures, and infected cells were observed in all parasite-treated samples (Figure 1a). In the parasite-treated cells, higher cell densities were recorded at 25 °C compared to the 13 °C treatment. The parasite infection rate was higher at the lower temperature (Figure 1b). In sum, the increased temperature does not eliminate the ability of the parasite, *L. coscinodisci,* to infect its host. However, it can influence the host, *C. granii,* and its ability to resist parasite infection.

### 2.2. UHPLC-HRMS Analysis Reveals Altered Metabolome Patterns Associated with Different Temperatures and Parasite Infection in Diatom Cells

To better understand the metabolic response in *C. granii* and provide insights into how temperature and parasitic infection can affect cellular processes, we profiled the cell extracts from cultures grown for 7 days with and without parasite (experiment 2). First, 40 mL cultures from the above experiments were filtered, and the recovered alga cells were extracted with methanol. The extracts were dried and taken up with 80 μL of methanol before UHPLC-HRMS analysis. Chromatographic separation using C18 and ZIC-HILIC columns yielded two data matrices of 1210 and 1195 features, respectively. The features are ions, which are characterized by a retention time, mass to charge (*m*/*z*) value, and peak intensity, and were annotated with a chemical formula based on spectral isotopic pattern analysis with Compound Discover (Thermo Fisher Scientific, Bremen, Germany). These features represented the ion traces of molecules obtained during UHPLC-HRMS analysis. After normalization based on the diatom cell counts, intensities were log-transformed and pareto-scaled in MetaboAnalyst (URL: https://www.metaboanalyst.ca/, accessed on 28 February 2024). The results of the UHPLC-HRMS analysis are shown in Figure 2, and we showed that cellular extracts are significantly discriminated depending on the temperature and the parasite infection. Principal component analysis (PCA) was conducted to reveal metabolic differences between algal cells grown at 13 °C and 25 °C and those treated or not with the parasite *L. coscinodisci* (NP: no parasite, WP: with parasite) (Figure 2). Significant discrimination in cellular metabolic profiles was observed, with a total explained variance of 79.8% and 68.8% for the profiling with C18 and ZIC-HILIC columns, respectively (Figure 2a,b). Metabolite variation driven by PC1 was related to higher temperature, while PC2 variation was associated with parasite treatment for the C18 dataset (Figure 2a). A few features were associated with parasite treatment, which could be explained by the low infection rates, which never reached over 20% (Figure 1b). A significant difference was thus mainly identified by algal metabolites up-regulated in cells grown at higher temperatures. Significant metabolites that drove these PCA were further examined using volcano plot analysis (Appendix A), and those that characterized cells grown in higher temperatures, regardless of the parasite treatment, were further identified using MS/MS analysis. Volcano plots were also inspected to search for and select significant features (Appendix A).

Spectral similarity matching of the significant features was conducted with public databases using GNPS, and fragmentation analysis was performed using SIRIUS:CSIFinger ID (Appendix A). UHPLC-HRMS analysis using the ZIC-HILIC column facilitated the identification of a diverse range of polar and hydrophilic compounds (Appendix A), such as serine, cysteinoleic acid, and dimethylsulfoniopropionate (DMSP), which were all significantly up-regulated in algal cells grown at 25 °C (Figure 3a). These structures were confirmed with analytical standards (Appendix A). Furthermore, chemicals associated with fatty acid metabolism were also identified as more abundant in cells grown at 25 °C, e.g., carnitine, acetylcarnitine, and eicosapentaenoic acid, and were confirmed with standards (Figure 3b). In the ZIC-HILIC dataset, two metabolites, ectoine and the recently described 2-homoectoine, were found and annotated based on spectral similarity matching of their MS/MS fragments with those of the public library. These two metabolites were not significantly regulated at 13 °C but downregulated in parasite-treated cells compared to untreated algae at 25 °C (Figure 3d).

Utilizing reverse-phase chromatography with a C18-column, we successfully identified two metabolites that discriminated between untreated algae grown at 13 °C and those treated with the parasite *L. coscinodisci* (Appendix A). First was eicosapentaenoic acid, detected in positive polarity as *m*/*z* 303.2327 for [M+H]^+^ at a retention time of 8.2 min; second was lumichrome, a riboflavin-derived metabolite, detected as *m*/*z* 243.0887 for [M+H]^+^ at 3.7 min. Both metabolites were confirmed by co-injection studies with standards (Appendix A). Eicosapentaenoic acid (EPA) was detected in both parasite-treated and untreated cell metabolomes but with significantly higher abundance in untreated cells at both temperatures (Figure 3b). Eicosapentaenoic acid can underpin a modulation in lipid metabolism during parasitism [22,23]. Here, levels of EPA were significantly higher at 25 °C but decreased in both temperature conditions under parasite treatment. Lumichrome was also significantly down-regulated in parasite-treated cells and was generally up-regulated in cells grown at 25 °C (Figure 3c).

## 3. Discussion

### 3.1. Effect of Temperature on Cell Abundance of Bloom-Forming Diatom C. granii and Parasite Infectivity of L. coscinodisci

The bloom-forming diatom *Coscinodiscus granii* is an essential primary producer that inhabits the oceans. It is thereby subjected to environmental changes. Here, we recorded the effect of temperature on the algal cell abundance, cell metabolome, and parasite infectivity. A high cell density was recorded for *C. granii* grown at 25 °C compared to the 13 °C treatment, indicating optimum growth at higher temperatures. This was particularly observed in the exponential phase between 6 and 12 days of incubation and the late stationary phase after 25 days of incubation (Appendix A). The cell density only remained higher at 13 °C in the late exponential phase. Previous studies have demonstrated that both *C. granii* and *L. coscinodisci* can be found in waters at a temperature range of 13 °C to 20 °C, with the percentage of infected cells ranging from 7.1% to 41.9% [24,25], which aligns with our present findings in laboratory experiments conducted at 13 °C. Also, it was shown that aquatic parasite abundance might change according to temperature and the host’s susceptibility to infection [9,15,26], and it would be interesting to test the effect of higher temperature on a broad strain spectrum of *L. coscinodisci* and other parasites targeting diatoms.

We report that the parasitic oomycete *L. coscinodisci* can infect the diatom *C. granii*, regardless of the incubation temperature (Figure 1a,b). The infection rate was lower in algal cultures grown at 25 °C compared to the 13 °C treatment (Figure 1b). Therefore, *C. granii* might bloom with higher cell abundance, and the parasitic oomycete might be less efficient in infecting the algal population at higher temperatures. The number of reproductive parasite cells can also vary depending on the virulence of the parasite strains employed, which can influence the total cell density of cultures [27]. The number of late infected algal cells (sporocysts) by the parasite *L. coscinodisci* never reached above 30% in laboratory experiments when cultured at 13 °C and was even lower when cultured at 25 °C (Figure 1b), which also follows the usual rates observed in the field during algal blooms [28]. The infection rates observed in our cultures are consistent with data from field observations that record that *L. coscinodisci* is present in natural blooms of *Coscinodiscus* at an infection rate of 5 to 47% [20]. We postulate that temperature may be just one factor among other abiotic factors that influence parasite infectivity, and algal resistance might be regulated by other means or factors.

### 3.2. Effect of Temperature on Cell Metabolic Profiles of the Bloom-Forming Diatom C. granii in Interaction with the Parasitic Oomycete

Diatoms exhibit adaptations to varying environmental temperatures, including adjustments in gene expression, membrane structure, and cellular metabolism [29]. Our metabolomics study provides some insights into understanding the metabolic response in *C. granii* according to the incubation temperature and parasite infection. We further observed key metabolic changes associated with temperature change that parasite exposure also impacted. Multiple studies have demonstrated the influence of abiotic factors such as temperature on host–symbiont interactions. For instance, in marine macroalgae *Delisea pulchra*, host–pathogen interactions are temperature-dependent, with temperature-regulated production of furanones preventing infection from pathogenic bacteria *Ruegeria* sp. [30]. Similarly, studies on *Micromonas* sp. have shown that the temperature can alter the viral lytic and lysogeny life cycle strategies of Prasinoviruses, potentially impacting ocean biogeochemistry [31]. Few compounds unique to the parasite-treated cells were detected in the present study, but the low infection rate might prevent the detection of subtle changes in the metabolome (Figure 1b).

Several algal metabolites were identified in the quality control pool sample with standards, including metabolites involved in sulfur metabolism, fatty acid metabolism, and riboflavin-derived substances (Figure 3, Appendix A). Reports have demonstrated that rising temperatures increase phytoplankton’s DMSP and cysteinolic acid levels [32], which aligns with our findings on *C. granii* (Figure 3a). DMSP is produced in parasite-infected diatoms, although in lesser amounts than in non-infected cells (Figure 3), and might relate to *C. granii* susceptibility to *Lagenisma* oomycete. DMSP has been found as a marker of *Parvilucifera* parasites infecting the dinoflagellate *Alexandrium minutum* [17], and when hydrolyzed into dimethylsulfide, it can enable parasite reproduction [33]. Although detected in lesser amounts in *Lagenisma*-treated diatom cells than in healthy cells (Figure 3), DMSP might also play a role in diatom–oomycete interaction that has yet to be demonstrated. The role of DMSP in marine symbiotic interactions is well-documented. For example, in the relationship between the microalga *E. huxleyi* and the algicidal bacterium *Roseobacter* sp., the alga produces DMSP, which the microbe metabolizes and releases methanethiol in return [34]. Algal hosts with higher DMSP levels tend to be more susceptible to infection [34], potentially due to the attraction properties of this compound and its associated metabolites that are released [35]. Cysteinolic acid is a sulfur-containing metabolite abundant in multiple phytoplankton species and is an adaptive strategy for high salinity stress [36]. This compound was detected as significantly abundant in untreated and parasite-treated cells grown at 25 °C and might have a role in alleviating the effect of temperature increase or during host–parasite interaction. Ectoine and derivatives, including the recently newly described 2-homoectoine, are produced by marine algae and bacteria, and their abundance can increase in cells during salinity stress [37]. Here, parasite-treated cells display a reduced amount of ectoine, and no significant chances of an increase in incubation temperature were observed. This suggests that ectoine and derivatives might play a role in host–parasite interactions but not in response to temperature increases.

Among the significantly up-regulated metabolites in cells grown at 25 °C are three chemicals—we highlighted carnitine, acetylcarnitine, and EPA. In eukaryotes and microalgae, acetylcarnitines produce acetyl-CoA in plastids, which serve as a source of acetyl-CoA for fatty acid synthesis in the plastid [38]. Carnitine is essential for the transfer of fatty acids across the inner mitochondrial membrane and is also an osmolyte involved in the response to salinity stress in plants and microalgae [39], but its role in temperature changes and diatom–parasite interaction is not yet known. Here, we observed an increase in carnitine and acetylcarnitine with higher incubation temperatures, regardless of the parasite treatment. It is also detected in all parasite-treated cultures (Figure 3b). In plants, fungal pathogenesis relies heavily on fatty acid metabolism, acetyl-CoA generation, and glyoxylate metabolism in the peroxisome, which is essential for pathogenesis, and this process, which depends on carnitine, may be conserved across terrestrial and aquatic habitats [40]. Eicosapentaenoic acid is a major fatty acid in diatoms and a precursor to many bioactive oxylipins [23]. Microalgae synthesize EPA primarily through aerobic metabolic pathways [41]. Nitrogen levels significantly influence EPA content. While low nitrogen boosts overall lipid production, the percentage of EPA decreases [42]. Nitrogen-replete conditions favor the production of EPA and polyunsaturated fatty acids in microalgae [43,44]. It has been theorized that high cell densities and cellular turnover could favor higher intracellular levels of EPA to maintain the membrane integrity of organelles in microalgae [45]. A slight increase in EPA levels during elevated temperatures could be tied to the enhanced cell growth rate observed for *C. granii*, especially in the early growth phase (Appendix A). Intracellular parasites may also depend on the host’s central carbon metabolism and lipids for fueling their development, replication, and propagation [46]. Additionally, EPA can be transformed into polyunsaturated aldehydes (PUA) and other oxylipins that are not covered by our method but could potentially contribute to algal defense against parasites [46]. This might support our observations, indicating that parasite-treated cells of diatom *C. granii* are characterized by decreased levels of EPA.

Finally, lumichrome, a riboflavin derivative, was detected in *C. granii* cells and was significantly up-regulated in both parasite-treated and untreated cells incubated at 25 °C (Figure 3c). It was also detected in parasite-infected cells, although in lesser amounts. Here, we first report this compound’s association with marine parasitic oomycete-infected diatom cultures and algae subjected to temperature increase. This compound plays a vital role in plant development [47], bacterial quorum sensing [48], and larval metamorphosis in ascidian *Halocynthia roretzi* [49]. The photodegradation of riboflavin can also produce it [50], and it has been found in the exometabolomes of microalgae such as *Phaeodactylum tricornutum*, *Chlamydomonas* sp., and *Desmodesmus* sp. [51]. Previous studies involving microalgae and lumichrome focus on its production by algae growth-promoting bacteria and not on the algal production of lumichrome [52]. Multiple studies also demonstrate that lumichrome supports cellular photosynthetic activity and has an overall positive effect on the growth of microalgal species such as *Phaeodactylum tricornutum*, *Chlorella sorokiniana* [52], and *Auxenochlorella protothecoides* [53]. Supporting these studies, we also suggest that high lumichrome levels correlate with high cell density and growth in diatoms, particularly at higher incubation temperatures, regardless of parasite treatment. However, further studies on *C. granii* must address whether the enhanced growth at elevated temperatures can be tied to lumichrome or if this compound has a role in parasite infection. No studies so far discuss the impact of temperature and interaction with parasites in lumichrome levels. A previous study using bacteria suggests that secreted lumichrome levels decline with a temperature rise and are maximum at a low temperature of 10 °C [54]. However, the role of lumichrome in other marine microbes and the impact of biotic interaction with parasites herein has not been studied thus far. Further studies are needed to clarify lumichrome’s role in parasite infection.

## 4. Conclusions

In this study, we show how temperature influences the cell abundance and metabolism of the bloom-forming diatom *C. granii* and the infectivity of its parasite *L. coscinodisci*. The relative abundance of several metabolites can be tied to elevated temperatures, switching the up-regulation of metabolites involved in sulfur metabolism and fatty acid-related metabolism. We also report that metabolites of ectoines decrease during parasite infection but increase at higher temperatures. We highlight that changing temperature induces a specific metabolic rewiring in algal cells during parasite infection. The findings may have broader implications for the ecology of marine environments, especially regarding the balance between organisms, the role of parasites in ecosystems, and the factors influencing algal bloom dynamics in a warming environment.

## 5. Materials and Methods

### 5.1. Strains and Culture Conditions for the Biological Experiments

The diatom strain RCC7046 and the parasitoid oomycete *L. coscinodisci* strain LagC19 were isolated from the same sample originating from an algal bloom at Helgoland in November 2019. The parasitoid oomycete strain was maintained in its diatom host by constant reinoculation into exponentially growing host cells every 15 days since its sampling in 2019. The diatom *C. granii* was deposited at the Roscoff Culture Collection under the strain number RCC7046 (URL: https://roscoff-culture-collection.org/ (accessed on 10 February 2023). *L. coscinodisci* and *C. granii* strains are maintained in a continuous culture by constantly propagating parasites into healthy host cultures and are available upon request. All cultures were maintained at 13 °C under a light intensity of 100 mE m^2^ s^−1^ following a 14 h: 10 h light/dark cycle. Cultures were grown in 40 mL Tissue Flasks (Sarsted) containing artificial seawater medium (ASW) [55].

### 5.2. Culture Conditions for the Biology Experiments

Initial inoculation from stock cultures was conducted using *C. granii* cells in the late exponential phase (initial cell density of 20 cells mL^−1^) and was used for the algal growth experiment. Culture flasks in biological triplicates were incubated at 25 °C or 13 °C. Cell concentration was recorded at ten-time points over 30 days of incubation (Appendix A). Cell density was determined by counting 1 mL of culture in a Sedgwick–Rafter chamber (Pyser-SGI).

To obtain a parasite suspension containing *L. coscinodisci* zoospores, 9-day-old infected *C. granii* cultures grown at 13 °C were filtered through a 20 μm cell strainer (PluriSelect), and the filtrate was used to inoculate in healthy cultures. For the infection experiment, 100 µL of a 9-day-old culture infected with *L. coscinodisci* (infection rate of 30%, cell density of 160 cells mL^−1^) was inoculated in 40 mL of healthy cultures (cell density of 200 cells mL^−1^). Cell density and infected cells were recorded by counting cells in 1 mL in the Sedgwick–Rafter chamber (Marienfeld, Germany). The number of healthy cells, total cell number, and number of infected cells (late infection stage sporangium) were determined under a light microscope (Leica DM 2000, Leica Microsystems, Wetzlar, Germany) in a Sedgwick–Rafter chamber (Pyser-SGI). Cultures were incubated under the two temperature regimes, 13 or 25 °C, and cells were counted and inspected every two days. Images were taken using an inverted microscope (AE30, Motic, Meyer, Houston, TX, USA) for regular checks and sub-cultivation, and a Zeiss Imager 2 (Carl Zeiss, Jena, Germany) was used to record microscopic images. This experiment was also conducted with biological triplicates. Raw data are presented in Appendix A. Statistical analysis and visualization of cell counts were conducted in GraphPad Prism Version 10.

### 5.3. Metabolic Extractions of Diatom Cultures

We followed protocols outlined in [16] for cell extraction and analysis using comparative metabolomics (5.3, 5.4, and 5.5). After seven days of incubation of the infection experiment, cultures were filtered under reduced pressure on GF/C microfiber filters (Whatman plc, Maidstone, UK) deposited on filter plates (Ø 24 mm, pore size of 4, glass edge, VWR International GmbH, Radnor, PA, USA) in a filtration unit (Duran/pp, VWR International GmbH, Radnor, PA, USA). The blank sample consisted of 40 mL of ASW medium without microbes. The GF/C filters were transferred to 2 mL safe-lock Eppendorf tubes and extracted with 1.5 mL of methanol (99.8%, anhydrous, SIGMA-ALDRICH Chemie GmbH, Munich, Germany). Cells were disrupted by ultrasonication in an ultrasonic cleaner Emmi-D280 (EMAG-AG, Herford, Germany) for 10 min at room temperature. The resulting mixture underwent centrifugation at 16,000× *g* at 4 °C for 15 min, facilitating debris sedimentation. Supernatants (1.2 mL) were carefully transferred to new tubes and subjected to further centrifugation for 20 min at 12,000× *g*. The resulting supernatants were then transferred into 1.5 mL glass vials, and the solvent was evaporated using a desiccator. The dried samples were stored at −20 °C until UHPLC-HRMS (Thermo Fischer Scientific, Bremen, Germany) analysis. Reconstitution of the dried extracts was performed with 80 μL of methanol. The resulting solution was transferred into 1.5 mL safe-lock Eppendorf tubes and underwent centrifugation at 16,000× *g* at 4 °C, as mentioned above, for 20 min. Subsequently, 50 μL of the resulting supernatant was transferred into LCMS glass vials containing glass inserts for subsequent analysis. To prepare the Quality Control (QC) pooled sample, 5 μL of each sample (excluding blanks) was pooled into a QC pool mix.

### 5.4. Chromatography

We followed protocols outlined in [16] for UHPLC-HRMS analysis and MS data acquisition. Samples for LC-MS analysis were measured with a Dionex UltiMate 3000^®^ (Dionex, Germering, Germany) coupled with a Q-Exactive Plus Orbitrap mass spectrometer (Thermo Scientific, Bremen, Germany). Two distinct chromatographic protocols were implemented to identify a wide range of metabolites, from non-polar to polar, using C18 and ZIC-HILIC columns.

An Accucore C18 column (100 × 2.1 mm, particle size of 2.6 μm, Thermo Scientific Fisher, Dreieich, Germany) was employed for the chromatographic separation of non-polar metabolites. Phase (A) consisted of UHPLC-grade water (2% acetonitrile, 0.1% formic acid), and phase (B) consisted of UHPLC-grade acetonitrile (0.1% formic acid). The separation process was initiated with 100% (A) at a flow rate of 0.4 mL min^−1^; a steadily increasing gradient of (B) for 8 min to reach 100% (B), which was held for 3 min; and finishing with 100% of (A) for 1 min. The sample injection volume was 10 μL.

For the chromatographic separation of polar metabolites, a SeQuant ZIC-HILIC column (150 × 2.1 mm, particle size of 5 µm, pore size of 200 Å, Merck, Darmstadt, Germany) was used, complemented with a SeQuant ZIC-HILIC guard column (20 × 2.1 mm, Merck, Darmstadt, Germany). The phases consisted of (A) UHPLC-grade water (2% acetonitrile, 0.1% formic acid) and (B) UHPLC-grade acetonitrile (10% water, 1 mmol L^−1^ of ammonium acetate). The separation commenced with 100% (B) at a flow rate of 0.6 mL min^−1^ for 1 min, steadily increasing the gradient of (A) for 6.5 min until 25% (B) was obtained, followed by 100% B for 3.5 min and finishing with 85% (B) for 2 min. The sample injection volume used was 1 μL.

### 5.5. Mass Spectrometry Analysis and Data Transformation

Mass spectrometry was carried out on a Q-Exactive Plus Orbitrap (Thermo Fisher Scientific, Dreieich, Germany). Analytes were monitored in full MS scan (Resolution 70,000, AGC target 3 × 10^6^, Maximum IT of 200 ms, with polarity switch). The scan range extended from *m*/*z* 75 to 1125, with a total runtime of 12 min (C18 column) or 10 min (ZIC-HILIC column).

Data-dependent MS2 (resolution of 17,500, AGC target of 1 × 10^5^, maximum IT of 50 ms, loop count of 5, Top N5, and isolation window of 0.4 *m*/*z*) with a three-stepped collision energy of 15, 30, and 45 was carried out on the QC pool sample to enable metabolite identification and annotation. The MS2 measurements were acquired separately in positive and negative modes. Total Ion Chromatograms and Extracted Ion Chromatograms were vizualized in QualBrowser (Thermo Fisher Scientific, Bremen, Germany) (Appendix A)

Raw data were processed using Compound Discoverer version^TM^ software (v 3.3.2.31; Thermo Fisher Scientific, Bremen, Germany) for data deconvolution and metabolite identification. The software was used to detect chromatographic peaks, perform retention time alignment using QC samples, detect unknown compounds, and group compounds across samples to generate a list of features. Each feature is characterized by a *m*/*z* value, retention time (min), and chemical formula annotated based on the isotopic pattern from the HRMS mass spectrum. For compound detection and fragment identification, the mass tolerances set for MS identification was 5 ppm, and the minimum peak intensity was 1 × 10^6^. The chromatographic S/N Threshold was set to 1.5 for peak detection without baseline removal. For grouping compounds and MS identification, mass tolerance was 10 ppm, with a retention time tolerance of 1 min and an S/N threshold at 5 (for Gap filling). The relative standard deviation value was set at 50, excluding features not represented in all pool samples (QC) replicates. The compound list was exported as a .csv file; the intensities were normalized based on cell density count (using the Scale Area node on Compound Discoverer) and analyzed with MetaboAnalyst 5.0. (URL: https://www.metaboanalyst.ca/ accessed on 12 December 2023). The compound list was exported as a .csv file, and the intensities were normalized based on cell density count and analyzed with MetaboAnalyst 5.0. The cell-based normalized intensities were further log-transformed and Pareto-scaled. In MetaboAnalyst, the PCA loading plots, volcano plots, and pattern hunter analysis were used to mine significant metabolites that discriminated cell metabolic profiles from cultures grown at 25 °C or 13 °C or treated with parasites. Of 1195 features from the ZIC-HILIC dataset of *C. granii* treated by temperature and parasite, 478 were excluded based on the filter using the Interquartile range. Of 1210 features from the C18 dataset of *C. granii* treated using temperature and parasite, 484 were excluded based on the filter using the Interquartile range. Significant features that were selected for further identification are shown in Appendix A.

### 5.6. Metabolite Identification

Significantly regulated metabolites identified using the statistical analysis were further investigated by acquiring MS/MS spectra using ddMS experiments on QC pool samples (Appendix A). For the identification and confirmation of significant features, the MS/MS spectra of selected ions were compared using SIRIUS (v5.8.3) and CSI: FingerID, as well as those of analytical standards. Depending on the tree fragmentation score and the percentage returned using CSI:FingerID tool, putative identification of unknowns was performed. MS/MS raw spectra are available in the files tab. GNPS was also used to identify features not previously identified using SIRIUS and confirm the identity of the zwitterions and polar metabolites recovered from the ZIC-HILIC and C18 analysis. Assignment of identity with a confidence level was conducted using a threshold of 90% similarity obtained in SIRIUS and a minimal cosine score of 0.7 in GNPS. Analytical standards were measured to confirm the metabolites lumichrome and eicosapentaenoic acid recovered from the C18 analysis. Furthermore, DMSP, cysteinoleic acid, serine, carnitine, and acetylcarnitine were confirmed with co-injection of the QC pool sample with analytical standards. Ectoine and 2-homoectoine were identified based on spectral similarity matching with MS2 from GNPS database (URL: http://gnps.ucsd.edu accessed on 1 March 2024) [56]. All MS/MS spectra of our reference standards were deposited and publicly available online in the GNPS spectral libraries. The charts were made using the Metabolomics Spectrum Resolver. All spectral mirror charts had a cosine similarity score of not less than 70. Identical MS/MS spectra proved that the identity of compounds from *C. granii* was identical to the reference standards. Significant metabolites that were identified can be found in Appendix A. Co-injection studies of lumichrome and eicosapentaenoic acid with QC algal pool samples are displayed in Appendix A.

## Figures and Tables

**Figure 1 plants-13-03415-f001:**
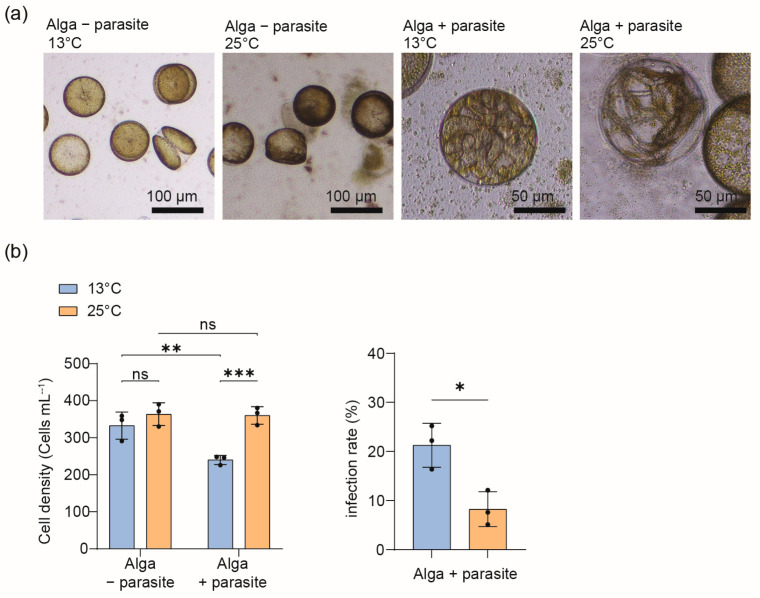
Temperature impacts on diatom host abundance and parasite infectivity. (**a**) Microscopy pictures of *C. granii* with (algae + parasite) and without parasite (algae − parasite) cultivated at 13 and 25 °C. Infection caused by the parasite *Lagenisma coscinodisci* was detected at both temperatures in *C. granii* cultures inoculated (alga + parasite). Infected cells harboring the parasite sporangia (late sporocyte phase) were observed at 13 °C and 25 °C. (**b**) Monitoring of cell density of infected cells and infection rate in algal cultures of *C. granii* grown at 13 °C and 25 °C and inoculated or not with the parasite *L. coscinodisci* (alga + parasite, or alga − parasite) (*N* = 3, the error bars denote the standard deviation). The statistical significance of cell abundance was tested with Two-way ANOVA, and a *t*-test was conducted to compare infected cell density and infection rate at different temperatures (*** *p* < 0.001, ** *p* < 0.01, * *p* < 0.05; ns, non-significant).

**Figure 2 plants-13-03415-f002:**
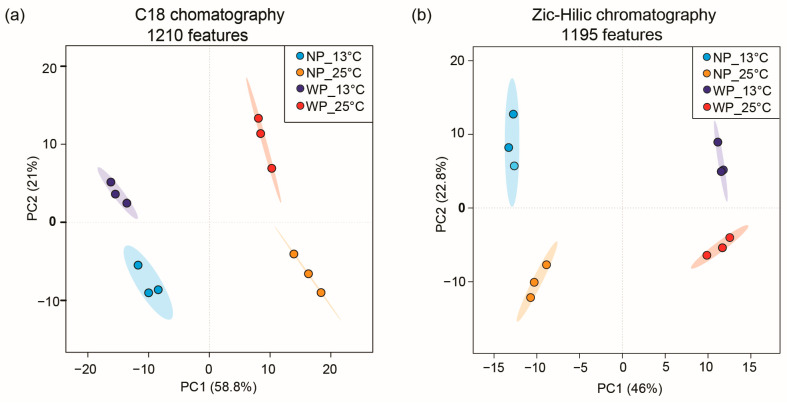
Temperature and parasite infection alter cellular metabolic profiles of the diatom *Coscinodiscus granii*. Principal component analysis of the features datamatrix obtained from UHPLC-HRMS analysis and data processing showed high discrimination between metabolic profiles of the diatom *C. granii* cells treated with parasites (WP) and untreated cells (NP) incubated at 13 °C and 25 °C for both (**a**) C18 and (**b**) ZIC-HILIC chromatography. The features are ion peak signals characterized by a chemical formula, *m*/*z* value, and retention time.

**Figure 3 plants-13-03415-f003:**
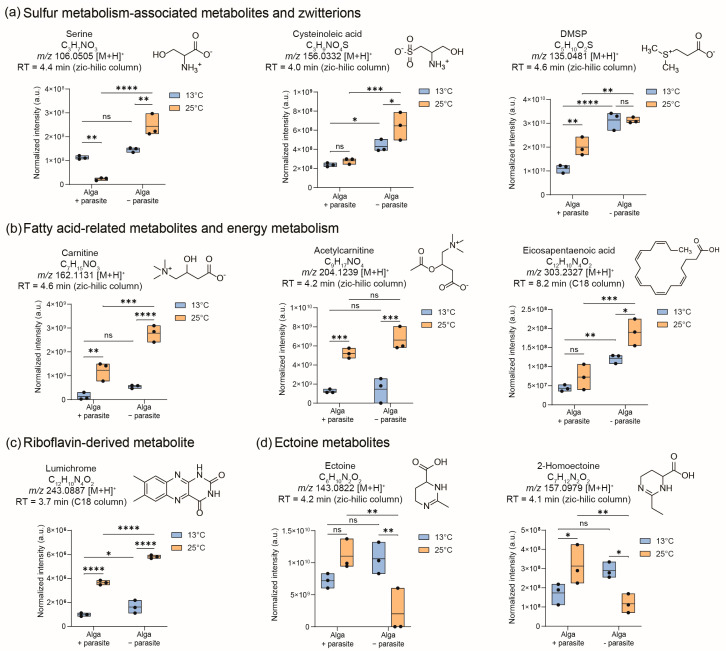
The identified compounds discriminated metabolic profiles of diatom *C. granii* cells treated with parasites (WP) from untreated samples (NP) incubated at 13 °C and 25 °C (*N* = 3, 3 biological replicates). Significant metabolites identified were from (**a**) metabolites related to sulfur metabolism and zwitterions, (**b**) metabolites involved in energy metabolism and related to fatty acids, (**c**) riboflavin-derived lumichrome, and (**d**) ectoine metabolites. The box plots display the maximum, minimum, and median lines; first/third quantiles; and the samples as points for the significant metabolites. Column bars indicate standards of error. Two-way ANOVA was conducted to achieve statistical significance using cell-based normalized intensities of the peaks (*p*-value: **** *p* < 0.0001, *** *p* < 0.001, ** *p* < 0.01, * *p* < 0.05; ns, non-significant).

## Data Availability

The microbial strains are maintained continuously and are available upon request. The datasets are available in the MassIVE spectral database under the dataset number MSV000095877, and the links are URL: https://massive.ucsd.edu/ProteoSAFe/dataset.jsp?task=a88ae0ffac114bde8d8a9bc65e5b698b (accessed on 15 October 2024); URL: ftp://MSV000095877@massive.ucsd.edu (accessed on 15 October 2024); URL: ftp://massive.ucsd.edu/v08/MSV000095877/ (accessed on 15 October 2024).

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
