# Peer review of "The Impact of Temperature on Host–Parasite Interactions and Metabolomic Profiles in the Marine Diatom *Coscinodiscus granii"

_plants, 2024, doi:10.3390/plants13233415_

Round 1

Reviewer 1 Report

Comments and Suggestions for Authors

The manuscript title “Impact of Temperature on Host-Parasite Interactions and Metabolomic Profiles in the Marine Diatom Coscinodiscus granii” is conducted well and has scientific worth. I have some minor suggestions to improve the current version of the manuscript.

Comments for authors are as follows:

1-      Figure 1 and 2: first give a common title to the figure than explain the subsection of the figures a, b, …

2-      In figures 1 and 3: What do the column bars means?? Is it St. error or St. deviation?? Please add the information in the figure legends.

3-      Line 395: “30,000 x g at 4°C for 15 minutes” the author did centrifuge at 30,000 x g? Please recheck and provide appropriate references for that.

4-      Line 399: “until UHPLC-HRMS.” Which company? Model?

5-      In methods section 5.3, 5.4, and 5.5 the authors didn’t add any references. Is all the procedure not referenced before?? I suggest the authors add the reference to the corresponding sections.

6-      The authors said they performed ANOVA; I suggest authors add the ANOVA table in the main text.

Author Response

We want to thank the reviewer for the highly constructive feedback and suggestions. We have thoroughly revised the manuscript after considering all the suggestions. Please see below our point-by-point reply.

  1. The titles were added in the caption, and the subsection descriptions were added.
  2. The bar denotes the standard deviation, and the information is added to the captions of figures 1 and 3. In Figure 3, the upper line of the box represents the highest reading obtained (peak area of sample for a given metabolite) and the lower line indicates the lowest reading.
  3. This has been corrected.
  4. UltiMate 3000 UHPLC Dionex, this coupled to a Q-Exactive Plus Orbitrap mass spectrometer (Thermo Fisher Scientific). This has been now added.
  5. Thank you for the comment. We added the reference we used for the protocols in the corresponding lines.
  6. The ANOVA table has been added in the supplementary section, As Table S3.

Reviewer 2 Report

Comments and Suggestions for Authors

In the study, the cold-water diatom Coscinodiscus grani is exposed to two unfavourable factors (elevated temperature and parasite infection). However, the results showed that an elevated temperature of 25°C is more favorable for cold-water diatoms. Despite this, the authors continue to consider it a stress factor. Moreover, parasite-treated cells exhibit reduced amounts of ectoin, DMSP, carnitine and acetylcarnitine, metabolites that are known to increase under stress.

 The authors do not explain (a) why the temperatures 13°C and 25°C were chosen for algae cultivation, (b) why 7-days culture was used for analysis, (c) what are these matrices (1210 and 1195) and why were they used to evaluate the metabolic profile?

 The authors cannot explain why the temperature of 25C is more favorable for the cold-water diatom Coscinodiscus grani than 13C.

 Authors did not show the results of UHPLC-HRMS analysis.

 This study does not provide any data on fatty acids in the diatom, so discussion of eicosapentaenoic acid is irrelevant.

 The significance and aim of the work were not written.

 The study was apparently performed only once and with a small number of repetitions (n=3). This makes it possible to doubt the correctness of the results obtained.

 This manuscript is filled with overclaims. This study does not provide any data on climate change. The authors should not overestimate the results of their experiments, assuming their importance for addressing climate change issues.

 The logic is broken especially in discussion section and the structure of the manuscript is not respected.

 Many parts of the manuscript do not correspond to the sections. Thus, the Results section describes the methodological details and arguments of the experiments. The Discussion contains a description of the results and introduction.

 Conclusions should not include a description of the results.

 There is a strong feeling that the authors have no experience in preparing papers for scientific journals Authors are encouraged to send the manuscript to senior scientists who can advise them.

  Following attached the point-to-point review comments:

 L. 49 Environmental conditions and abiotic factors are synonyms.

L. 54-56 “Specific populations of a given species can also perform better under parasite exposure than others at increasing temperatures, shaping the competition between aquatic organisms even after parasite decline”. What did you want to say? Rewrite it.

L. 72-83. It is for Abstract or Results. At the end of Introduction the goals and objectives of the study should be formulated.

Results

For experiments, cultures with an initial cell density of 200 cells mL-1 were used.

In the exponential growth phase (Day 16 and 18), cell density  was significantly higher in cultures grown at 13°C than at 25°C. It should be explained and discussed.

L. 90-94 is not for results.

Fig. 1a. does not carry information, it should be deleted or explained.

According to Fig. 1b, the parasite infection does not influence on cell density in culture at 25°C. It should be discussed.

L. 91-120. What age (days) was the culture measured?

L. 120-126 does not concern the Results.

Fig. 2. Is it Principal component analysis or UHPLC-HRMS analysis? Correct. PCA of what?

L. 192-197. It does not confirm the discussion but more suitable for introduction.

L. 221-224. Does not relevant to the results obtained.

L. 267-270. Does not relevant to the discussion since the coral Pocillopora is an animal.

L. 295-310. It should be deleted.

Author Response

In the study, the cold-water diatom Coscinodiscus granii is exposed to two unfavourable factors (elevated temperature and parasite infection). However, the results showed that an elevated temperature of 25°C is more favorable for cold-water diatoms. Despite this, the authors continue to consider it a stress factor. Moreover, parasite-treated cells exhibit reduced amounts of ectoine, DMSP, carnitine and acetylcarnitine, metabolites that are known to increase under stress.

We want to thank the reviewer for the highly constructive feedback and suggestions. We have thoroughly revised the manuscript after considering all the comments. Please see below our point-by-point reply.

We edited the lines that state that 25 °C is the less favorable temperature for cell growth. Indeed, the diatom also grows well at this temperature. We have corrected our initial assumption and indicated that we originally intended to test the effect of the two temperatures on parasite infectivity and host cell abundance. In figure 3, we show that ectoine, DMSP, carnitine and acetylcarnitine, are increased in cells cultivated at 25°C, indicating that the diatom strain C. granii RCC7046 can grow well and produce these metabolites at higher incubation temperature. The findings suggest that this diatom strain might well adapt to temperature change and thrive under warming conditions, we have thus modified the text accordingly.

The authors do not explain (a) why the temperatures 13°C and 25°C were chosen for algae cultivation, (b) why 7-days culture was used for analysis, (c) what are these matrices (1210 and 1195) and why were they used to evaluate the metabolic profile?

We hope to address them below and have also revised the manuscript accordingly.  

  • (a)The temperatures of 13°C and 25 °C were chosen because 29 °C is the highest temperature at which the diatom granii was found to occur naturally. The temperature of 13 °C is the routine condition that the parasite L. coscinodisci has successfully maintained in our laboratory since its isolation.
  • (b)We mention that 7-day cultures were used for the analysis in order to reduce the number of empty frustules (cell lysis caused by parasite infection) in the culture medium, which accumulates the more the infection progresses in the cell population. High parasites (zoospores) would be present at Day 9 of incubation as the majority of the cells would be either infected or dead, while at day 7 of incubation, the late infection stages (sporangia) were counted and also found to be significantly different in cultures grown at 25 °C compare with those cultured at 13 °C. Further edits were made in the corresponding text to explain this.
  • (c)Ions peak signals, or Features in metabolomics, were detected by UHPLC-HRMS analysis, and their intensities between samples were analyzed using a Principal Component Analysis (PCA). Further edits were made in the corresponding text and caption of the figure to indicate what these features are and that they are characterized by a chemical formula, m/z value, and retention time. The statistical analysis using a PCA on the UHPLC-HRMS data matrices obtained highlighted that the cell metabolome profiled is significantly different between samples derived from parasite-infected cultures and grown at the two temperatures.

 The authors cannot explain why the temperature of 25 °C is more favorable for the cold-water diatom Coscinodiscus grani than 13°C.

Our findings suggested that the high temperature may increase cellular metabolism and the rate of cell division. However, we would need other in-depth experiments using for example inhibitors to provide an insight into the mechanism supporting these observations, which was beyond our aims for this study. Our goal was to study the effect of temperature on host cell abundance, parasite infectivity, and cell metabolome. We modified the rationale in the introduction and reformulated our aims to clarify our objectives.

 Authors did not show the results of UHPLC-HRMS analysis.  This study does not provide any data on fatty acids in the diatom, so discussion of eicosapentaenoic acid is irrelevant.

The results of the UHPLC-HRMS analysis are represented in figures 2 and 3, including comparative metabolomics analysis of cellular metabolic profiles of cultures extracted and identification of significant features with their abundance plot. We identified a number of the significant metabolite (features) that explained the discrimination found between parasite-treated cultures grown at the two temperatures (Supplementary Table S2). While we did not target the whole range of fatty acids, we identified eicosapentaenoic acid using coelution with standard (supplementary figure S18) as one of the significant features. The other results of the UHPLC HRMS are shown in supplementary figures S2 to S16, with the volcano plot analysis of comparative analysis. We have now correctly mentioned these figures in the manuscript in lines 177-180.

 The significance and aim of the work were not written.

 The study was apparently performed only once and with a small number of repetitions (n=3). This makes it possible to doubt the correctness of the results obtained.

 This manuscript is filled with overclaims. This study does not provide any data on climate change. The authors should not overestimate the results of their experiments, assuming their importance for addressing climate change issues.

 The logic is broken especially in discussion section and the structure of the manuscript is not respected.

 Many parts of the manuscript do not correspond to the sections. Thus, the Results section describes the methodological details and arguments of the experiments. The Discussion contains a description of the results and introduction.

 Conclusions should not include a description of the results.

Thank you for your constructive feedback, we went through the whole manuscript, from abstract to conclusion, and excluded the parts that create confusion, including overclaims, we have also rewritten our goals in the introduction, and wrote our aims and significance. We also modified each section so that they follow the correct logic for the structure of the manuscript.  These modifications were made with the revision tools and are highlighted using tracked changes.

 There is a strong feeling that the authors have no experience in preparing papers for scientific journals Authors are encouraged to send the manuscript to senior scientists who can advise them.

  Following attached the point-to-point review comments:

L 49 Environmental conditions and abiotic factors are synonyms.

We strongly agree and the Line 49 has been rectified.

L 54-56 “Specific populations of a given species can also perform better under parasite exposure than others at increasing temperatures, shaping the competition between aquatic organisms even after parasite decline”. What did you want to say? Rewrite it.

The sentence has been rewritten 

L 72-83. It is for Abstract or Results. At the end of Introduction the goals and objectives of the study should be formulated.

The sentences were modified and added in lines 97- 100 and 85 to 90.

Results

For experiments, cultures with an initial cell density of 200 cells mL-1 were used.

In the exponential growth phase (Day 16 and 18), cell density was significantly higher in cultures grown at 13°C than at 25°C. It should be explained and discussed.

The growth curve measurement as mentioned in the Supplementary is another separate experiment from the infection experiments (carried out with 200 cells ml -1). For the infection experiments, cultures in the late exponential phase were taken and subcultured with fresh medium (acclimatized to the respective temperatures). This was done to make sure the infections proceeded with the same number of cells and that nutrients were no longer a limiting factor. We indicated now these in the text with tracked changes.

L 90-94 is not for results.

Fig. 1a. does not carry information, it should be deleted or explained.

According to Fig. 1b, the parasite infection does not influence on cell density in culture at 25°C. It should be discussed.

We added a better description for Fig. 1. Microscopy pictures indicate the late infection stage of the parasite in its diatom cells (Lagenisma sporangia), which was used for determination of the infectivity and describe that the parasite reproduction is occurring at both temperature. Thank you for the insightful observation in Fig.1b, we added comments in Lines 256-259 and modified the text accordingly.

L 91-120. What age (days) was the culture measured?

The age of the cultures prior to extraction is 7 days. After the initial seeding density of 200 cells per ml-1. We edited lines 97 – 127 accordingly, Thank you for the comment.

L 120-126 does not concern the Results.

Fig. 2. Is it Principal component analysis or UHPLC-HRMS analysis? Correct. PCA of what?

Thank you for the comment, the we edited accordingly Line  210 - 216

L 192-197. It does not confirm the discussion but more suitable for introduction.

L 221-224. Does not relevant to the results obtained.

L 267-270. Does not relevant to the discussion since the coral Pocillopora is an animal.

L 295-310. It should be deleted.

We deleted the non-relevant part and modified accordingly for the last four points.

Reviewer 3 Report

Comments and Suggestions for Authors

This is a nicely prepared manuscript, well written, and clearly organized. It should be a useful contribution to the literature. One very small suggestion, I believe that the proper format for some of the abbreviations should be i.e., not e.g., because the former represents 'that is' and the latter represents 'for example' (Lines 77 and 87).

Author Response

Thank you for your comments, we modified the text accordingly with the tracked changes.

Reviewer 4 Report

Comments and Suggestions for Authors

No further comments.

Author Response

Thank you for your appreciation and time.

Round 2

Reviewer 2 Report

Comments and Suggestions for Authors

After evaluating the changes made to the manuscript by the author, which were requested in the first review, I recommend the acceptance of this manuscript in Plants.